# Dynamic fingerprint of fractionalized excitations in single-crystalline $Cu_3Zn(OH)_6FBr$

Ying Fu[1,8], Miao-Ling Lin[2,3,8], Le Wang[1], Qiye Liu[1], Lianglong Huang[1], Wenrui Jiang[1], Zhanyang Hao[1], Cai Liu[1], Hu Zhang[4], Xingqiang Shi[4], Jun Zhang[2,3,5], Junfeng Dai[1], Dapeng Yu[1], Fei Ye[1,6], Patrick A. Lee[7], Ping-Heng Tan[2,3,5 ✉] & Jia-Wei Mei[1,6 ✉]

Beyond the absence of long-range magnetic orders, the most prominent feature of the elusive quantum spin liquid (QSL) state is the existence of fractionalized spin excitations, i.e., spinons. When the system orders, the spin-wave excitation appears as the bound state of the spinon-antispinon pair. Although scarcely reported, a direct comparison between similar compounds illustrates the evolution from spinon to magnon. Here, we perform the Raman scattering on single crystals of two quantum kagome antiferromagnets, of which one is the kagome QSL candidate $Cu_3Zn(OH)_6FBr$, and another is an antiferromagnetically ordered compound $EuCu_3(OH)_6Cl_3$. In $Cu_3Zn(OH)_6FBr$, we identify a unique one spinon-antispinon pair component in the $E_{2g}$ magnetic Raman continuum, providing strong evidence for deconfined spinon excitations. In contrast, a sharp magnon peak emerges from the one-pair spinon continuum in the $E_g$ magnetic Raman response once $EuCu_3(OH)_6Cl_3$ undergoes the antiferromagnetic order transition. From the comparative Raman studies, we can regard the magnon mode as the spinon-antispinon bound state, and the spinon confinement drives the magnetic ordering.

[1] Shenzhen Institute for Quantum Science and Engineering, and Department of Physics, Southern University of Science and Technology, Shenzhen, China. [2] State Key Laboratory of Superlattices and Microstructures, Institute of Semiconductors, Chinese Academy of Sciences, Beijing, China. [3] Center of Materials Science and Optoelectronics Engineering & CAS Center of Excellence in Topological Quantum Computation, University of Chinese Academy of Sciences, Beijing, China. [4] College of Physics Science and Technology, Hebei University, Baoding, China. [5] Beijing Academy of Quantum Information Science, Beijing, China. [6] Shenzhen Key Laboratory of Advanced Quantum Functional Materials and Devices, Southern University of Science and Technology, Shenzhen, China. [7] Department of Physics, Massachusetts Institute of Technology, Cambridge, MA, USA. [8] These authors contributed equally: Ying Fu, Miao-Ling Lin. ✉email: phtan@semi.ac.cn; meijw@sustech.edu.cn

Quantum spin liquid (QSL) represents a new class of condensed matter states characterized by the long-range many-body entanglement of topological orders[1–9]. The lattice of the spin-1/2 kagome network is a long-sought platform for antiferromagnetically interacting spins to host a QSL ground state[10–16]. However, a structurally ideal realization of the kagome lattice in experiments is rare. Herbersmithite [ZnCu$_3$(OH)$_6$Cl$_2$] is the first promising kagome QSL candidate[3,16–23], in which no long-range magnetic order was detected down to low temperature[17,18], and inelastic neutron scattering revealed a magnetic continuum, as a hallmark of fractionalized spin excitations[20,22]. Up to date, most, if not all, experimental information on the nature of kagome QSL relies on a single compound of Herbertsmithite. Considering the fact that a lattice distortion has recently been confirmed in Herbersmithite[24,25], which stimulates investigations on the subtle magneto-elastic effect in the kagome materials[26,27], an alternative realization of the QSL compound with the ideal kagome lattice is still in urgent need. Zn-Barlowite [Cu$_3$Zn(OH)$_6$FBr] is another candidate for a kagome QSL ground state[28–38] with no lattice distortion being reported yet. Measurements on the powder samples didn't detect the long-range magnetic order down to temperatures of 0.02 K, four orders of magnitude lower than the Curie–Weiss temperature[30,32]. Besides the lack of magnetic order, the fractionalized spin excitations, i.e., spinons, is essential evidence for the long-range entanglement pattern in QSL. However, spectroscopic evidence for the deconfined spinon excitations in Zn-Barlowite is still lacking, in part due to unavailable single-crystal samples.

Raman scattering is sensitive to the local symmetries depending on the light polarization[39,40], and also capable of detecting magnetic excitations ranging from the spin-wave magnon excitation to deconfined spinons[41–50]. Raman scattering has previously been reported for Herbertsmithite and revealed the multiple spinon scattering process[19]. In recent years, the atacamite family ReCu$_3$(OH)$_6$Cl$_3$ (Re=Y, Eu, Sm, and Nd) with the perfect kagome lattice has been synthesized and a chiral 120° antiferromagnetic (AFM) order with the wave vector **q** = 0 is identified in the ground state[51–55]. The kagome spin systems can be described by the kagome Heisenberg model with the Dzyaloshinski–Moriya (DM) interaction

$$H = J\sum_{\langle ij\rangle}(\mathbf{S}_i \cdot \mathbf{S}_j) + D\hat{z}\cdot\sum_{\langle ij\rangle}\mathbf{S}_i\times\mathbf{S}_j, \qquad (1)$$

where summation runs over nearest-neighbor bonds $\langle ij\rangle$, and $J$ and $D$ are the nearest-neighbor exchange and the DM interaction constants, respectively, for the spins $S_{i,j}$ on the $i$- and $j$-th sites. We ignore the in-plane DM interactions regarding to the previous electron paramagnetic resonance measurements in the related kagome systems[55,56]. A DM interaction larger than the critical value of $(D/J)_c \sim 0.08$ induces a chiral 120° AFM order from the QSL state[57–59]. By the first-principle calculations (Supplementary Note 1), Zn-Barlowite and EuCu$_3$(OH)$_6$Cl$_3$ have $D/J$ values of 0.05 and 0.3, resulting in QSL and AFM ground states, respectively, consistent with the experimental identification of the ground states[30,54]. While the elementary spin excitation of the kagome QSL is the deconfined spinon, the low energy excitation in the kagome AFM ordered states is the magnon. A direct comparison by the magnetic Raman scattering can reveal the evolution from deconfined spinons in Zn-Barlowite to magnons in EuCu$_3$(OH)$_6$Cl$_3$, but has not been performed yet.

In this work, we exclude the kagome lattice distortion by angle-resolved polarized Raman (ARPR) scattering and second-harmonic-generation (SHG), and reveal the spin dynamics of spinon excitations on the single-crystalline Cu$_{3.18}$Zn$_{0.82}$(OH)$_6$FBr. We observe a remarkable $E_{2g}$ magnetic Raman continuum, which

can be decomposed into one spinon–antispinon pair (one-pair (1P)) and two spinon–antispinon pair (two-pair (2P)) components of spinon excitations, in line with theoretical studies of the kagome QSL[60]. The one-pair continuum is unique, serving as the fingerprint of spinons. In a control experiment, beside the two-magnon (2M) magnetic Raman continuum, we probe a sharp one-magnon (1M) Raman peak in EuCu$_3$(OH)$_6$Cl$_3$ below the AFM transition temperature. The magnon peak emerges from the 1P continuum in the magnetic Raman scattering, can be regarded as the bound state of the spinon–antispinon excitations. As schematically summarized in Fig. 1, our comparative Raman study demonstrates the spinon deconfinement and confinement in the kagome QSL compound and ordered antiferromagnet, respectively. The AFM order transition can be thought to be driven by the spinon confinement.

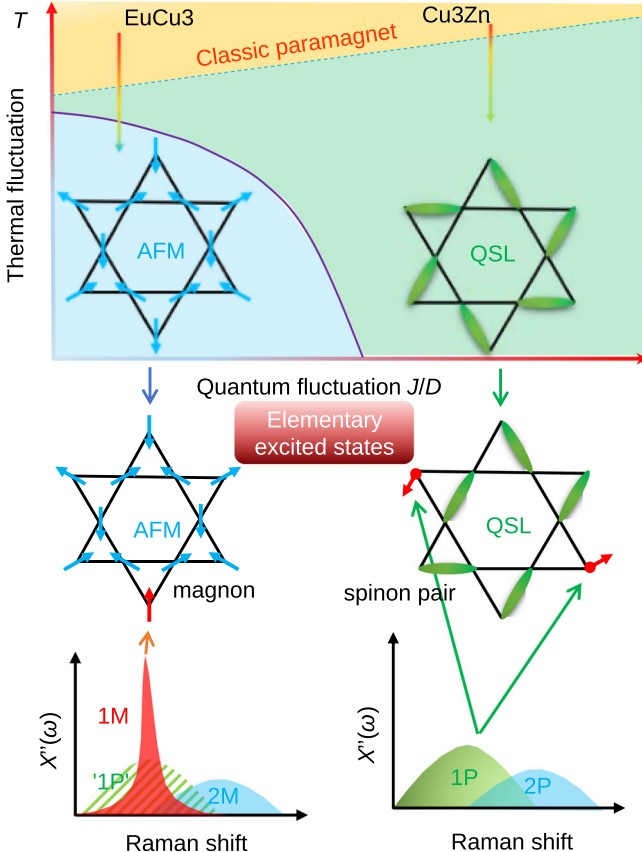

**Fig. 1 Schematical comparative Raman responses for the AFM and QSL states.** With a large DM interaction $D$, the kagome antiferromagnet develops a chiral 120° AFM ground state. Increasing $J/D$, the fluctuation of the kagome system increases, driving the system into the QSL state. By increasing the temperature, the thermal fluctuation melts the magnetic order and turns the system into the classic paramagnetic state at high temperatures. Cu$_3$Zn and EuCu$_3$ have the QSL and AFM ground states, and allow spinon and magnon excitations, respectively. Magnetic Raman scattering measures different elementary excited states in the two different ground states. Here 1P and 2P denote the one-pair and two-pair spinon excitations, respectively. 1M and 2M in AFM ordered state denote the one- and two-magnon excitations, respectively. The 1M Raman peak in AFM measures the magnon while the 1P Raman continuum in QSL probes the spinon excitations. The shadow background of the 1M peak, marked as `1P', denotes the continuum above $T_N$ in EuCu$_3$, mimicking the 1P continuum in the QSL state.

## Results

We grown single crystals of Barlowite $Cu_4(OH)_6FBr$, Zn-Barlowite $Cu_{3.18}Zn_{0.82}(OH)_6FBr$, and $EuCu_3(OH)_6Cl_3$ (we use the short-hand notation $Cu_4$, $Cu_3Zn$, and $EuCu_3$, respectively) with high quality ("Methods" and Supplementary Note 2). The interlayer $Cu^{2+}$ concentration (18%) is comparable to that (15%) in Herbertsmithite[61]. We estimate the superexchange strength for the kagome spins in $Cu_3Zn$ as $J \simeq 13$ meV by the Curie–Weiss temperature $\Theta_{CW} = -220$ K (Supplementary Note 2)[62]. The superexchange interaction for $EuCu_3$ is about $J \simeq 7$ meV[53–55]. Note the electronic ground state of $Eu^{3+}$ in $EuCu_3$ is the non-magnetic $^7F_0$ configuration.

Figure 2a presents the temperature evolution of Raman spectra in $Cu_3Zn$ with sharp phonon modes superimposing on the magnetic continuum background. With the help of first-principles calculations, we assign the symmetry representations for phonon modes in Supplementary Note 3. No structural phase transition is observed in $Cu_3Zn$ down to 4 K. We tracked the Raman spectral evolution of the crystal structures from $Cu_4$ to $Cu_3Zn$ (Supplementary Note 4). $Cu_3Zn$ has no Raman-active mode related to the kagome $Cu^{2+}$ vibrations, indicating the kagome layer remains intact. $Cu_4$ has distorted kagome layers at 200 K, signaled by an extra phonon mode at $62$ cm$^{-1}$ corresponding to the kagome $Cu^{2+}$ vibration. The previous SHG study revealed the parity symmetry in Barlowite 2 $[Cu_4(OH)_6FBr]$ and Zn-Barlowite $[Cu_{3.66}Zn_{0.33}(OH)_6FBr]$[25]. We confirmed the inversion symmetry by SHG in our single crystals of $Cu_3Zn$ (Supplementary Note 6).

Figures 2b–d are the ARPR responses of $Cu_3Zn$ in three different polarization configurations ("Methods" and Supplementary Note 5). In the XX (XY) configuration, the incoming and outgoing light polarizations are parallel (perpendicular) and rotated simultaneously. In the X-only configuration, the outgoing polarization is fixed and only incoming light is rotated. Theoretically, the Raman cross section of a Kagome QSL ground state does not depend on the polarization of the incoming or outgoing light[39] and keeps invariant against rotating light polarization in the XX, XY, and X-only configurations. Figure 2b is the ARPR response for the magnetic continua at low frequency with the integrated Raman susceptibility $\chi' = \frac{2}{\pi} \int_{10cm^{-1}}^{60cm^{-1}} \frac{\chi''(\omega)}{\omega} d\omega$, where the susceptibility is related to the Raman intensity $I(\omega) = (1 + n(\omega))\chi''(\omega)$ with the bosonic temperature factor $n(\omega)$. Figure 2c and d are the corresponding results of the $Br^- E_{2g}$ phonon, and $O^{2-} A_{1g}$ phonon modes, respectively. For threefold rotation symmetry, the $A_{1g}$ mode response follows the $\cos^2(\theta)$ function of the rotation angle $\theta$ in X-only configuration, keeps constant in XX polarization, and vanishes in XY configuration; the $E_{2g}$ mode is isotropic in all the three configurations. The magnetic continuum contains both $A_{1g}$ and $E_{2g}$ channels at high temperature (290 K), and only the $E_{2g}$ channel at low temperature (4 K). The experimental ARPR responses agree well with the theoretical dash-dotted curves, confirming the threefold rotational symmetry in the magnetic excitations (Fig. 2b) and lattice vibrations (Figs. 2c, d). We notice that in Herbertsmithite, although it was not discussed, the lattice distortion was evident by the anisotropic ARPR responses[19] and may account for the difference from our results.

Having established the structurally ideal realization of the kagome lattice by SHG and ARPR scattering, and the absence of the thermodynamic anomaly, we now present our spectroscopic results of spin dynamics in $Cu_3Zn$ with subtracting phonon contributions. Figure 3a–c are the magnetic continuum of $Cu_3Zn$ in the $A_{1g}$ channel, which is activated only at high temperatures, and disappears at low temperatures. The integrated Raman susceptibility in Fig. 3b fits the thermally activated function, $\chi'(T) \propto e^{-\omega^*/T}$ with $\omega^* = 53$ cm$^{-1}$. The result suggests the $A_{1g}$ continuum measures the thermal fluctuation of the interacting kagome

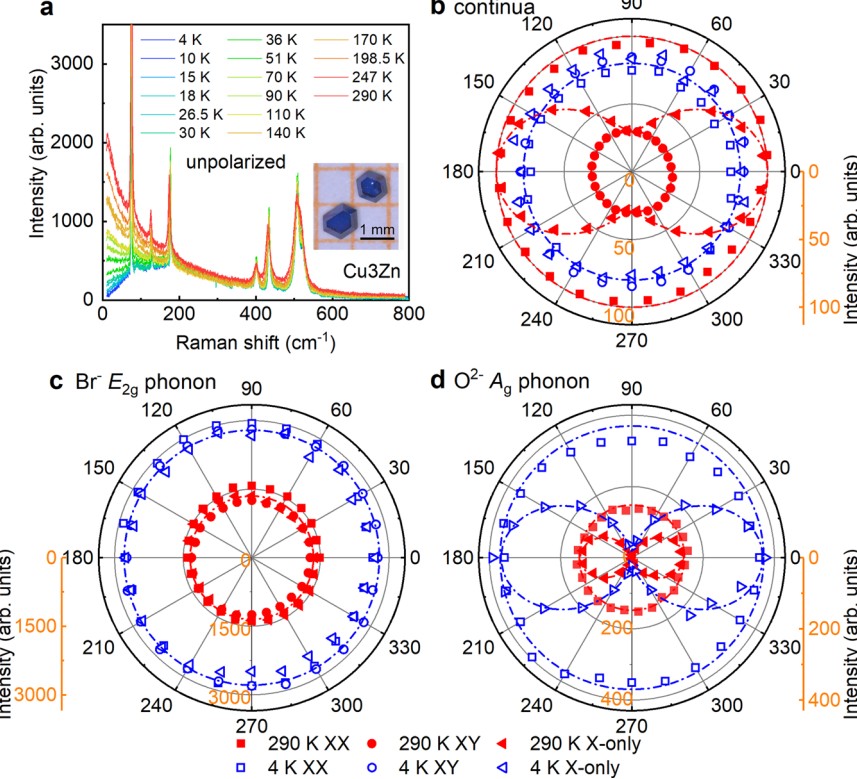

**Fig. 2 Temperature dependent and ARPR spectra in $Cu_3Zn$. a** Temperature evolution of unpolarized Raman spectra in $Cu_3Zn$. The inset is the photo of single crystals. ARPR intensity for low-energy continua (**b**), the $Br^- E_{2g}$ phonon (75 cm$^{-1}$) (**c**), and the $O^{2-} A_{1g}$ phonon (429 cm$^{-1}$) (**d**). The dash-dotted lines are the corresponding theoretical curves based on the $C_3$ rotation symmetry.

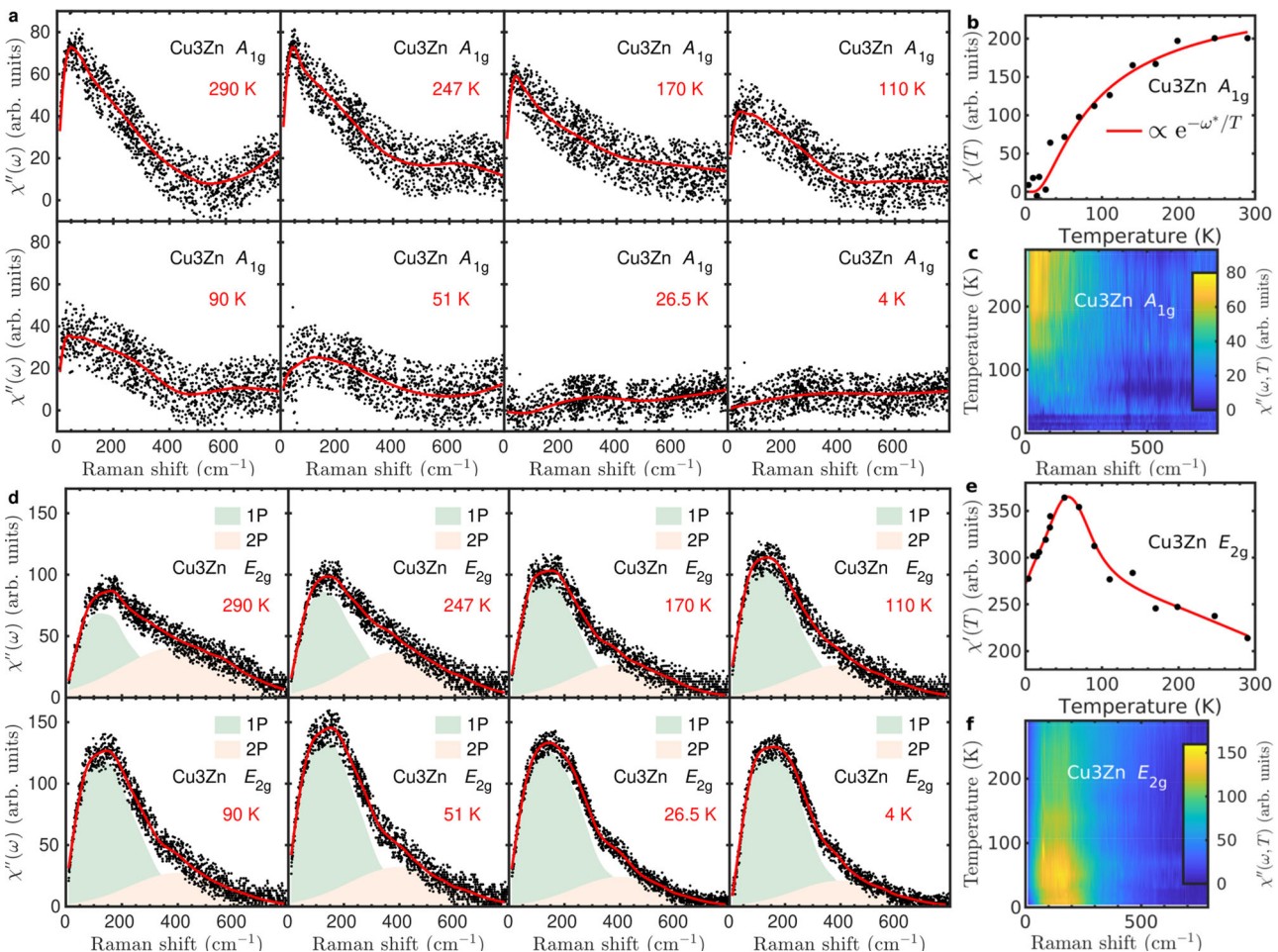

**Fig. 3 Temperature dependent magnetic Raman continua in Cu₃Zn. a** The $A_{1g}$ Raman susceptibility $\chi''_{A_{1g}} = \chi''_{XX} - \chi''_{XY}$. The solid lines are guides to the eye.

**b** Temperature dependence of the $A_{1g}$ static Raman susceptibility $\chi'_{A_{1g}}(T) = \frac{2}{\pi}\int_{10\,cm^{-1}}^{400\,cm^{-1}} \frac{\chi''_{A_{1g}}(\omega)}{\omega}d\omega$. The solid line is a thermally activated function. **c** Color map of $\chi''_{A_{1g}}(\omega, T)$. **d** The $E_{2g}$ Raman response function $\chi''_{E_{2g}} = \chi''_{XY}$. The solid lines are guides to the eye. The light green and pink shadow marked as "1P" and "2P" represent the one-pair and two-pair components of Raman continuum. **e** Temperature dependence of the $E_{2g}$ static Raman susceptibility $\chi'_{E_{2g}}(T) = \frac{2}{\pi}\int_{10\,cm^{-1}}^{780\,cm^{-1}} \frac{\chi''_{E_{2g}}(\omega)}{\omega}d\omega$. The solid line is a guide to the eye. **f** Color map of $\chi''_{E_{2g}}(\omega, T)$.

spins[41,63,64]. Different from the $A_{1g}$ channel, the pronounced $E_{2g}$ magnetic Raman continuum persists down to 4 K (Fig. 3d–f), indicating the intrinsic quantum fluctuation of the kagome spins. The substantial low energy component has a non-monotonic temperature dependence. It increases with the temperature decreasing from 290 K to 50 K, but decreases with further temperature reducing as shown in Fig. 3d–f. The $E_{2g}$ magnetic Raman susceptibility $\chi''(\omega, T)$ distributes the main spectral weight among the frequency region less than 400 cm⁻¹, and reaches the maximum at around 150 cm⁻¹ and 50 K, as shown in Fig. 3f.

The low-energy $E_{2g}$ Raman continuum is crucial as it has an origin of the spinon excitation in the kagome QSL from the theoretical perspective[60]. In the XY configuration for the $E_{2g}$ channel, the Raman tensor on the kagome lattice is written in terms of spin-pair operators[39,60,65,66]

$$\tau_R \propto \sum_R \mathbf{S}_{R3} \cdot (\mathbf{S}_{R1} + \mathbf{S}_{R+\mathbf{a}_21} - \mathbf{S}_{R2} - \mathbf{S}_{R-\mathbf{a}_1+\mathbf{a}_22}), \quad (2)$$

where $\mathbf{S}_{R1,2,3}$ are spin operators on three sites of the $R$-th kagome unit cell and $\mathbf{a}_{1,2}$ are the lattice vectors. The spin operator has the spinon $f_{i\sigma}$ representation $S_i^\alpha = \sum_{\sigma\sigma'} f_{i\sigma}^\dagger \tau_{\sigma\sigma'}^\alpha f_{i\sigma'}/2$ where $\tau^\alpha$ is the $\alpha$-th Pauli matrix. The spin-pair is $\mathbf{S}_i \cdot \mathbf{S}_j = -\frac{1}{2}\hat{\chi}_{ij}^\dagger \hat{\chi}_{ij}$ with $\hat{\chi}_{ij} = \sum_\sigma f_{i\sigma}^\dagger f_{j\sigma}$. In the mean field theory, the spinon hopping

amplitude $\chi = \langle \hat{\chi}_{ij} \rangle$ is non-zero. So we have 1P and 2P components in the Raman tensor[60]

$$\tau_R^{1P} \propto \chi \sum_R (\hat{\chi}_{R3,R1} + \text{h.c.}) + \cdots, \quad (3)$$

$$\tau_R^{2P} \propto \sum_R \hat{\chi}_{R3,R1}^\dagger \hat{\chi}_{R3,R1} + \cdots, \quad (4)$$

where $\cdots$ denotes omitted terms in Eq. (2) for the notation simplicity. While the 2P component is analogous to the 2M scattering, the 1P contribution is a unique prediction for spinon excitations in the kagome QSL. In Fig. 3d, we schematically decompose the $E_{2g}$ Raman continuum into 1P and 2P components of spinon–antispinon excitations. The 1P component has the maximum at 150 cm⁻¹ (1.4$J$), and extends up to 400 cm⁻¹ (3.8$J$) at low temperatures. The 2P component has the maximum at 400 cm⁻¹ (3.8$J$) and the cut-off around 750 cm⁻¹ (6.7$J$). The mentioned features (maxima and cut-offs) of 1P and 2P excitations in the $E_{2g}$ Raman response agree well with the theoretical prediction for the kagome QSL state[60].

In more detail, the 1P component dominates the $E_{2g}$ magnetic Raman continuum at low frequency. It displays the power-law behavior up to 70 cm⁻¹, with a significantly nonmonotonic temperature dependence, as shown in Fig. 4. The low-energy

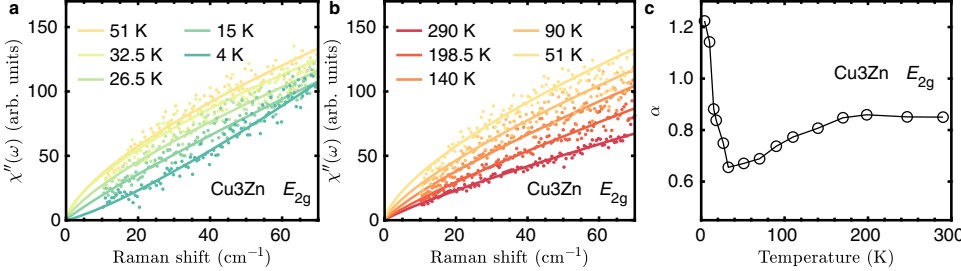

**Fig. 4 Power-law behavior for $E_{2g}$ magnetic Raman continua at low frequency in Cu₃Zn. a**, **b** are power-law fitting of $\chi''_{E_{2g}}(\omega) \propto \omega^\alpha$ at low and high temperatures, respectively. **c** Temperature-dependent exponent $\alpha$ for the power-law fitting.

continuum evolves from a sublinear behavior $T^\alpha$ with $\alpha < 1$ to a superlinear one $T^\alpha$ with $\alpha > 1$ as reducing the temperature. A central question for the kagome QSL is whether a spin gap exists. Previous results on the powder samples of Cu₃Zn suggest a small spin gap[30,32]. If such a gap exists, the power-law behavior of the $E_{2g}$ magnetic Raman continua sets an upper bound for the spin gap of 2 meV.

The theoretical calculation for kagome Dirac spin liquid (DSL) predicts the power-law behavior for the Raman susceptibility in the $E_{2g}$ channel at low frequency[60]. The 1P spinon excitation in DSL gives the linear density of state (DOS) $\mathcal{D}_{1P} \propto \omega$. The matrix element turns out to be exactly zero for all 1P excitations with $\omega = 0$ in the mean field Dirac Hamiltonian. As a result a Raman spectrum that scales as $\omega^3$ was predicted. However, the vanishing of the matrix element is somewhat accidental and depends on the assumption of a DSL in an ideal kagome Heisenberg model. Any deviation from the ideal DSL state, e.g., a small gap in the ground state[30,32], DM interactions, or other effects of perturbations[26,67], changes the wave functions and may result in a constant matrix element. In that case, the Raman spectrum will be simply proportional to the DOS of the 1P component $\mathcal{D}_{1P}$ which is linear in $\omega$. From our fitting for Cu₃Zn in Fig. 4, we find that $\alpha = 1.3$ when approaching zero temperature. The existence of a small gap in the spinon spectrum may explain the discrepancy.

Considering the interlayer $Cu^{2+}$ concentration (18%) in Cu₃Zn, we make a remark here about the disorder effect in the magnetic Raman scattering. The temperature-dependent $E_{2g}$ static magnetic susceptibility $\chi'_{E_{2g}}(T)$ of Cu₃Zn in Fig. 3 exhibit the maximal spin fluctuations at 50 K. The non-monotonic $T$-dependence deviating from the Curie–Weiss behavior is associated with the enhancement of nearest-neighbor spin correlations at low temperatures[67]. However, such significant deviation from Curie–Weiss behavior is masked by the interlayer $Cu^{2+}$ moments in the bulk thermodynamic measurements, e.g., heat capacity and bulk magnetization[30]. In contrast to a significant energy dependent magnetic Raman susceptibility $\chi''_{E_{2g}}(\omega)$ at 4 K in Cu₃Zn, the scattered neutron signal $\chi''_{INS}(\omega)$ in Herbertsmithite is overall insensitive to energy transfer, rather flat above 1.5 meV, but increases significantly with reducing energy below 1.5 meV due to the interlayer $Cu^{2+}$ ions[20,22]. So Raman scattering singles out the kagome magnetic excitations and remains unmasked in the presence of the interlayer $Cu^{2+}$ due to the matrix element effect as explained below. The Raman scattering measures the nearest-neighbor spin-pair $\tau_R \propto \mathbf{S}_i \cdot \mathbf{S}_j$ dynamics, but the spin pairs associated with the interlayer $Cu^{2+}$ ions are weaker than the singlet pairs for the kagome spins. As the light polarization in our Raman measurements is in the kagome $ab$ plane, and the projected factor of the spin-pairs associated with the interlayer $Cu^{2+}$ ions, $(\mathbf{r}_{ij} \cdot \hat{\mathbf{e}}_{in})(\mathbf{r}_{ij} \cdot \hat{\mathbf{e}}_{out})$, is small, as the related pair bond vector $\mathbf{r}_{ij}$ has the angle around 52° with respect to the kagome plane. As a result, the interlayer $Cu^{2+}$ ions contribute a negligible Raman matrix element and we ignore their effect in the discussions about

the Raman experiments. Moreover, the inelastic neutron scattering in Herbertsmithite measures the magnetic continuum up to $3J$[20], the same energy range as the 1P Raman component in Cu₃Zn. These results suggest that the magnetic Raman continuum originates from the kagome spins, and the 1P component has an origin of spinon excitations.

Figure 5 presents a control Raman study on the magnetic ordered kagome antiferromagnet EuCu₃, which has the antiferromagnetic superexchange strength $J \simeq 7$ meV. In Supplementary Note 7, the ARPR scattering on EuCu₃ confirms the threefold rotational symmetry. Above the ordering temperature $T_N = 17$ K, the magnetic Raman continuum in the $E_g$ channel displays the extended continuum, similar to the $E_{2g}$ magnetic continuum at 4 K in Cu₃Zn. Below $T_N$, a sharp peak, i.e., 1M peak as discussed below, is observed on top of the magnetic continuum. The integrated Raman susceptibility $\chi'_{E_g}(T)$ monotonically increases as lowering the temperature as shown in Fig. 5b, different from non-monotonic behavior in $\chi'_{E_{2g}}(T)$ of Cu₃Zn in Fig. 3e. The magnetic Raman susceptibility $\chi''(\omega, T)$ in EuCu₃ distributes the main spectral weight among the frequency region less than 400 cm⁻¹, and the magnon peak locates at 72 cm⁻¹ below 17 K, as shown in Fig. 5c.

To directly compare the 1P spinon continuum in Cu₃Zn and the 1M peak in EuCu₃, we plot the $E_g$ Raman response in EuCu₃ at selected temperatures in Fig. 6. The $E_{2g}$ Raman continuum in Cu₃Zn at 4 K is also plotted with the proper scale for the Raman frequency. Above $T_N = 17$ K, EuCu₃ has the substantial magnetic continuum with the profile similar to that in Cu₃Zn at 4 K. There are less pronounced low-energy continuum excitations in EuCu₃ than those in Cu₃Zn, probably due to the large DM interaction which suppresses the low-energy quantum fluctuations. Below $T_N$, a sharp magnon peak at 72 cm⁻¹ appears in EuCu₃ with the corresponding energy scale of the 1P continuum maximum in Cu₃Zn. We stress that the magnon Raman peak is direct spectroscopic evidence for the $\mathbf{q} = 0$ 120° non-collinear AFM spin configurations, and invisible in the $\sqrt{3} \times \sqrt{3}$ structure of the 120° AFM ("Methods").

For the AFM order state, the low-energy excitation is the spin-wave magnon which can be described in the spin-wave theory[68]. In the local spin basis $\tilde{\mathbf{S}}_i$ of the AFM order, we have the Raman tensors in the XY configuration of the $E_g$ channel for 1M and 2M components as following

$$\tau_R^{1M} \propto \sum_R (\tilde{\mathbf{S}}_{R1}^y + \tilde{\mathbf{S}}_{R2}^y - \tilde{\mathbf{S}}_{R3}^y), \qquad (5)$$

$$\tau_R^{2M} \propto \sum_R \tilde{\mathbf{S}}_{R3} \odot (\tilde{\mathbf{S}}_{R1} + \tilde{\mathbf{S}}_{R+\mathbf{a}_1 1} - \tilde{\mathbf{S}}_{R2} - \tilde{\mathbf{S}}_{R-\mathbf{a}_1+\mathbf{a}_2 2}), \qquad (6)$$

with the 2M spin-pair operator $\tilde{\mathbf{S}}_i \odot \tilde{\mathbf{S}}_j = \tilde{S}_i^x \tilde{S}_j^x + (\tilde{S}_i^y \tilde{S}_j^y + \tilde{S}_i^z \tilde{S}_j^z)/2$. For the details, please refer to the "Methods" section. Therefore, the $E_g$ Raman scattering in the AFM order state measures 1M and 2M excitations as demonstrated in Fig. 1. Thus, the magnon

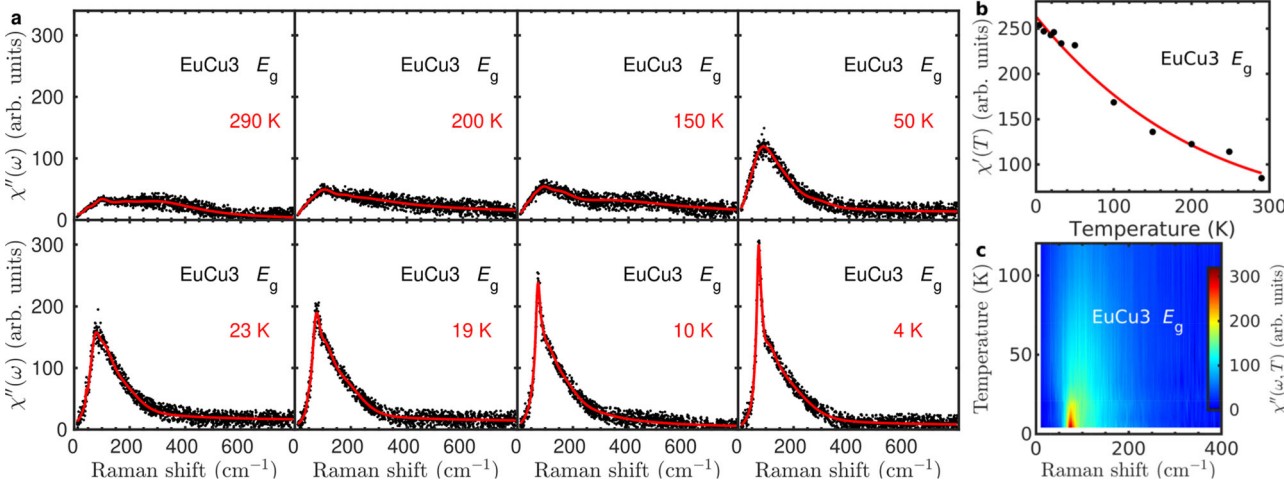

**Fig. 5 Temperature dependent $E_g$ magnetic Raman continua in EuCu$_3$. a** The $E_g$ Raman susceptibility $\chi''_{E_g} = \chi''_{XY}$. The solid lines are guides to the eye. A sharp magnon peak appears in the $E_g$ magnetic Raman continuum below the magnetic transition temperature $T_N = 17$ K. **b** Temperature dependence of the static Raman susceptibility in the $E_g$ channel $\chi'_{E_g}(T) = \frac{2}{\pi}\int_{10\,cm^{-1}}^{780\,cm^{-1}}\frac{\chi''_{E_g}(\omega,T)}{\omega}d\omega$. The solid line is a guide to the eye. **c** Color map of $\chi''_{E_g}(\omega,T)$.

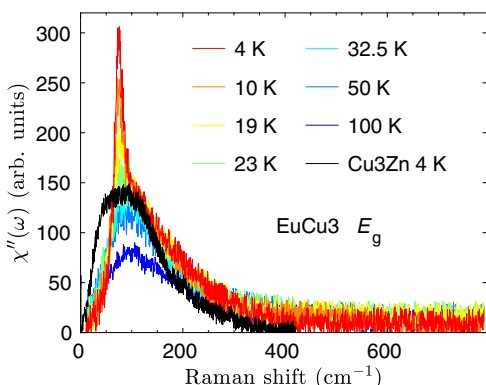

**Fig. 6 Comparative Raman studies of EuCu$_3$ and Cu$_3$Zn.** We select the $E_g$ magnetic Raman continua in EuCu$_3$ at several temperatures. For a comparison, we also plot the $E_{2g}$ magnetic Raman continuum in Cu$_3$Zn at 4 K with the Raman shift scaled by the superexchange energy ratio of 1.9.

excitation emerges from the 1P continuum and can be regarded as the bound state of the spinon–antispinon excitations.

## Discussion

Deconfined spinons yield to the magnetic continuum, however, the Raman continuum does not necessarily imply the spin fractionalization. Only 2M excitation itself gives rise to a Raman continuum in the ordered antiferromagnet[42]. In this work, the comparative Raman study in Cu$_3$Zn and EuCu$_3$ resolves this uncertainty. Guided by the theoretical prediction[60], the $E_{2g}$ Raman continuum can be decomposed into 1P and 2P components of the spinon–antispinon excitations. While the 2P component has the maximum at 3.8$J$, resembling the 2M broad peak[42], the 1P continuum in Raman is unique for QSL. Its maximum and extended range have the same energy scale as the spin-wave magnon peak in EuCu$_3$ and the inelastic neutron continuum cutoff (up to 3$J$) in the Herbertsmithite, respectively.

The 1P component of Raman continuum reveals fractionalized spin excitations, providing strong evidence for the kagome QSL ground state in Cu$_3$Zn. Our comparative Raman studies explicitly show the evolution from the deconfined spinon excitation in the kagome quantum spin liquid compound Cu$_3$Zn to the conventional magnon in the kagome ordered antiferromagnet EuCu$_3$.

On the material side, Zn-Barlowite is an ideal structural realization of the kagome lattice. Along with Herbertsmithite, the single-crystalline Zn-Barlowite stands able to single out the intrinsic properties of the kagome QSL.

## Methods

**Sample preparation and characterization**. High qualified single crystals of Zn-Barlowite was grown by a hydrothermal method similar to crystal growth of herbertsmithite[69,70]. CuO (0.6 g), ZnBr$_2$ (3 g), and NH$_4$F (0.5 g) and 18 ml deionized water were sealed in a quartz tube and heated between 200 °C and 140 °C by a two-zone furnace. After 3 months, we obtained millimeter-sized single crystal samples. The value of $x$ in Cu$_{4-x}$Zn$_x$(OH)$_6$FBr has been determined as 0.82 by Inductively Coupled Plasma-Atomic Emission Spectroscopy (ICP-AES). The single-crystal X-ray diffraction has been carried out at room temperature by using Cu source radiation ($\lambda = 1.54178$ Å) and solved by the Olex2.PC suite programs[71]. The structure and cell parameters of Cu$_{4-x}$Zn$_x$(OH)$_6$FBr are in coincidence with the previous report on polycrystalline samples[30,32]. For Barlowite(Cu$_4$(OH)$_6$FBr), the mixture of CuO (0.6 g), MgBr$_2$ (1.2 g), and NH$_4$F (0.5 g) was transferred into Teflon-lined autoclave with 10 ml water. The autoclave was heated up to 260 °C and cooled to 140 °C after 2 weeks. A similar growth condition to Barlowite was applied for the growth of EuCu$_3$(OH)$_6$Cl$_3$ with staring materials of EuCl$_3$·6H$_2$O (2 g) and CuO (0.6 g).

**Measurement methods**. Our thermodynamical measurements were carried out on the Physical Properties Measurement System (PPMS, Quantum Design) and the Magnetic Property Measurement System (MPMS3, Quantum Design).

The temperature-dependent Raman spectra are measured in a backscattering geometry using a home-modified Jobin-Yvon HR800 Raman system equipped with an electron-multiplying charged-coupled detector (CCD) and a ×50 objective with long working distance and numerical aperture of 0.45. The laser excitation wavelength is 514 nm from an Ar$^+$ laser. The laser-plasma lines are removed using a BragGrate bandpass filter (OptiGrate Corp.), while the Rayleigh line is suppressed using three BragGrate notch filters (BNFs) with an optical density 4 and a spectral bandwidth ~ 5–10 cm$^{-1}$ [72]. Thus, Raman signal down to 5 cm$^{-1}$ can be measured[73]. The 1800 lines/mm grating enables each CCD pixel to cover 0.6 cm$^{-1}$. The samples are cooled down to 30 K using a Montana cryostat system under a vacuum of 0.4 mTorr and down to 4 K using an attoDRY 1000 cryogenic system. All the measurements are performed with a laser power below 1 mW to avoid sample heating. The temperature is calibrated by the Stokes-anti-Stokes relation for the magnetic Raman continuum and phonon peaks. The intensities in two cryostat systems are matched by the Raman susceptibility. The ARPR measurements[40] with light polarized in the $ab$ kagome plane of samples were performed in parallel (XX), perpendicular (XY), and X-only polarization configurations (Supplementary Note 5).

SHG measurements were performed using a homemade confocal microscope in a backscattering geometry. A fundamental wave centered at 800 nm was used as excitation source, which was generated from a Ti-sapphire oscillator (Chameleon Ultra II) with an 80 MHz repetition frequency and a 150 fs pulse width. After passing through a ×50 objective, the pump beam was focused on the sample with a diameter of 2 µm. The scattering SHG signals at 400 nm were collected by the same objective and led to the entrance slit of a spectrometer equipped with a

thermoelectrically cooled CCD. Two shortpass filters were employed to cut the fundamental wave.

**Magnon Raman peak in kagome AFM ordered state**. With a large DM interaction $D$, the kagome antiferromagnet in Eq. (1) devolops a $\mathbf{q} = 0$ type 120° AFM order at low temperature in EuCu$_3$[53–55,57–59]. In terms of the local basis for the AFM order, we rewrite the Hamiltonian as

$$H = J\sum_{\langle ij\rangle}\tilde{\mathbf{S}}_i \odot \tilde{\mathbf{S}}_j + D\sum_{\langle ij\rangle}\tilde{\mathbf{S}}_i \otimes \tilde{\mathbf{S}}_j, \tag{7}$$

with

$$\tilde{\mathbf{S}}_i \odot \tilde{\mathbf{S}}_j = S_i^x S_j^x + \cos(\theta_{ij})(S_i^y S_j^y + S_i^z S_j^z) + \sin(\theta_{ij})(S_i^z S_j^y - S_i^y S_j^z), \tag{8}$$

$$\tilde{\mathbf{S}}_i \otimes \tilde{\mathbf{S}}_j = \sin(\theta_{ij})(S_i^y S_j^y + S_i^z S_j^z) + \cos(\theta_{ij})(S_i^y S_j^z - S_i^z S_j^y), \tag{9}$$

where $\theta_{ij}$ is an angle between two neighboring spins and $S_i^{x,y,z}$ below denotes the local basis of the AFM order. The effective linear spin wave Hamiltonian is given as

$$\mathcal{H}_{\text{eff}} = J\sum_{\langle ij\rangle}[S_i^x S_j^x + (\cos\theta_{ij} + \sin\theta_{ij}D/J)\times(S_i^y S_j^y + S_i^z S_j^z)], \tag{10}$$

for which the Holstein-Primakoff representation for spin operators in the local basis was applied and the energy dispersion was obtained in ref. [68].

In the local spin basis, we have the Raman tensor in the XY configuration is given as

$$\tau_R^{\text{XY}} = \frac{\sqrt{3}}{4}\sum_R \tilde{\mathbf{S}}_{R3} \odot (\tilde{\mathbf{S}}_{R1} + \tilde{\mathbf{S}}_{R+\mathbf{a}_2 1} - \tilde{\mathbf{S}}_{R2} - \tilde{\mathbf{S}}_{R-\mathbf{a}_1+\mathbf{a}_2 2}). \tag{11}$$

In the spin-pair operator $\tilde{\mathbf{S}}_i \odot \tilde{\mathbf{S}}_j$ in Eq. (8), there are two-magnon contribution in terms of $S_i^x S_j^x + \cos(\theta_{ij})(S_i^y S_j^y + S_i^z S_j^z)$, and one- and three-magnon contributions in terms of $\sin(\theta_{ij})(S_i^z S_j^y - S_i^y S_j^z)$. For the $\mathbf{q} = 0$ spin configuration, we find that $\tau_R^{\text{XY}}$ in Eq. (11) has the non-vanished one magnon contributions. For the $\sqrt{3}\times\sqrt{3}$ AFM state, $\tau_R^{\text{XY}}$ has no one-magnon contribution. Therefore, the observed one-magnon peak in the $E_g$ channel in EuCu$_3$ provides evidence for the $\mathbf{q} = 0$ spin ordering at low temperatures. In the linear spin-wave theory, we take $S^z$ in the local basis as a constant, $S_i^z = \langle S^z\rangle = 1/2$, and the Raman tensor in XY configuration is given as

$$\tau_R^{\text{XY}} = \frac{3}{8}\sum_R(S_{R1}^y + S_{R2}^y - 2S_{R3}^y), \tag{12}$$

in terms of the local basis, directly measuring the one magnon excitation.

For EuCu$_3$, the exchange interaction parameters are estimated as $J = 7$ meV, $D/J = 0.3$, leading to the magnon peak position of $\Delta_{sw} = 1.1J = 77$ cm$^{-1}$, very close to the measured value 72 cm$^{-1}$ in our Raman measurement of the one-magnon peak.

## Data availability

All data supporting the findings of this study are available from the corresponding authors upon reasonable request.

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

## Acknowledgements

This work was supported by the National Key Research and Development Program of China (2016YFA0301204), the program for Guangdong Introducing Innovative and Entrepreneurial Teams (No. 2017ZT07C062), by Shenzhen Key Laboratory of Advanced Quantum Functional Materials and Devices (No. ZDSYS20190902092905285), Guangdong Natural Science Foundation (No. 2020B1515120100) and by National Natural Science Foundation of China (Grant Nos. 11774143, 12004377 and 11874350), the CAS Key Research Program of Frontier Sciences (ZDBS-LY-SLH004) and China Postdoctoral Science Foundation (2019TQ0317 and 2020M682780). P.A. Lee acknowledges support by the US Department of Energy under grant number DE-FG02-03ER46076.

## Author contributions

J.W.M. conceived the project. P.H.T. conceived the experimental work of Raman spectroscopy. Y.F., L.W., L.H., W.J., and Z.H. synthesized single crystals of samples. M.L. and P.H.T. designed the Raman experiments. M.L., J.Z., and P.H.T. performed Raman measurements. Q.L. and J.D. performed the SHG measurements. Y.F., L.W., L.H., and C.L. performed and analyzed magnetic susceptibility and heat capacity measurements. H.Z., X.S., and J.W.M. performed first-principles calculations. J.W.M., Y.F., M.L., and P.H.T. analyzed the Raman data. P.A.L., J.W.M., and F.Y. worked on the theory. P.A.L., J.W.M., F.Y., P.H.T., and M.L.L. wrote the manuscript with contributions and comments from all authors.

## Competing interests

The authors declare no competing interests.
