## [Peer Review File · Nature Communications]

REVIEWER COMMENTS

Reviewer #1 (Remarks to the Author):

The authors report a detailed Raman study of the magnetic excitations in QSL candidate $\text{Cu}_3\text{Zn}(\text{OH})_6\text{FBr}$ and compare it with the ordered antiferromagnet $\text{EuCu}_3(\text{OH})_6\text{Cl}_3$. A broad feature is seen in the Raman susceptibility of both, which eventually appears to become a 1 magnon excitation in the AFM ordered system. The structure is also evaluated using Raman selection rules. The paper is generally very well written and is a nice result comparing the two systems to potentially show how spinons become confined. I am generally inclined to recommend publication, however, the paper could use some additional analysis to be at the level of nature communications. My specific comments are below:

1) The authors mention they use "X-only", but don't really explain why they chose to do so. It is not a common setup for Raman and thus it would be helpful to add a sentence. Similarly, was the incident or scattered light that was measured?

2) The authors mention observing a high T continuum in A_{1g} that disappears upon cooling. Could this be the quasi-elastic scattering often seen in such systems? If so does its temperature dependence follow the magnetic susceptibility of the sample?

3) In the Cu_3Zn the authors argue they observe the 1 pair and 2 pair spinon excitation. However, it is not really clear how they exclude 2 magnon excitations? In other words, do they have direct evidence these are from fractional excitations? For example, there is the recent work by the Burch group showing how to analyze the continuum to demonstrate the statistics are fermionic, have the authors done something similar or directly compared to theoretical calculations? (Note the authors do not cite the latest works on RuCl_3 from Choi and Burch, the former shows magnons emerging from the continuum in a magnetic field and the later how to analyze the spinon continuum).

Reviewer #2 (Remarks to the Author):

In this manuscript, the authors report Raman scattering data on single crystals of two quantum kagome antiferromagnets, of which one is the kagome QSL candidate $\text{Cu}_3\text{Zn}(\text{OH})_6\text{FBr}$ (short

notation Cu₃Zn), and another is an antiferromagnetically ordered compound EuCu₃(OH)₆Cl₃ (short notation EuCu₃).

The authors estimated the superexchange strength for the kagome spins in Cu₃Zn as $J=19\text{meV}$ and Curie-Weiss temperature $\Theta_{\text{CW}}=-220\text{K}$. The superexchange interaction for EuCu₃ is about $J=10\text{meV}$. They also established the structurally ideal realization of the kagome lattice of Cu₃Zn by SHG and angle-resolved Raman scattering.

At high temperatures, the two compounds show qualitatively the same Raman intensity, but at low temperatures the response is very different. A sharp magnon peak appears in EuCu₃ in the E_g magnetic Raman continuum below the magnetic transition temperature $T_N=17\text{K}$. In Cu₃Zn, at high temperature, the authors observe continuum in both E_{2g} and A_{1g} symmetry channels.

At low temperatures, only E_{2g} response remains, i.e. the authors see a remarkable E_{2g} magnetic Raman continuum, which differently from the response in A_{1g} channel persists down to 4 K, indicating the intrinsic quantum fluctuation of the kagome spins. No sharp peaks at low T is observed in Cu₃Zn, and a beautiful comparison of the Raman response in two compounds is presented in Fig.5. Then the authors claimed that it can be decomposed into one-pair and two-pair components of spinon excitations, in agreement with previous theoretical works.

These findings are interesting and deserve publishing.

However, I have some doubts about the interpretation of the

experimental data. The authors claimed that it can be decomposed into one-pair and two-pair components of spinon excitations, in agreement with previous theoretical works. They also discuss the implication of the power-law behavior in the Raman susceptibility in the E_{2g} channel at low frequency and suggest that the spinons might correspond to those in the Dirac kagome QSL. However, if a theory for the Raman response in the Dirac kagome QSL exists, why not to present it along with the experimental data, or at least perform a qualitative comparison? Also it is not clear for me if this theory can account for strongly non-monotonic T-dependence. The authors do not discuss this.

Also, in EuCu₃, the authors claimed that the magnon Raman peak is a direct spectroscopic evidence for the $q=0$ 120 degree non-collinear AFM spin configurations, but no comparison with theoretical results even on the level of the linear spin wave theory is presented.

I think the reader will benefit a lot if the authors address some of these questions prior to publication.

Reviewer #3 (Remarks to the Author):

The search for experimental realizations of quantum spin liquids has long been a daunting challenge. Kagome systems bear utmost promises with herbertsmithite $\text{ZnCu}_3(\text{OH})_6\text{Cl}_2$ being in the limelight for a decade. That said, circumstantial evidence casts looming doubts on its ground state's characteristics. Attempts at replacing Zn ions did not quench caveats. In the past few years, barlowite and its derivatives paved a way to replace the Cl in herbertsmithite. Strong frustration in both materials amplifies the effect of minimal perturbations in their structures, such that stepping away from herbertsmithite yet remaining in the spin-liquid regime is all the more desirable. Pioneer studies on Zn-barlowite show strong support of a spin-liquid ground state. More direct observations of spinons are obviously the imperatives.

The authors show spinon-pair excitation using Raman spectroscopy conducted on a single crystal sample. Similar observation was one of the inspiring evidence in herbertsmithite for which slight structural distortion has been reported recently. The authors carefully demonstrate the robustness of structure in Zn-barlowite, subject to the resolution of Raman spectroscopy, before presenting magnetic excitation which resemble those observed in herbertsmithite. The spectral profile is consistent with theoretical predictions on a Dirac $U(1)$ gapless spin-liquid state overall. In addition, the availability of $\text{EuCu}_3(\text{OH})_6\text{Cl}_3$ ---a resembling compound with an antiferromagnetic order below 17K---allows the authors to contrast magnon pairs vs spinon pairs. The distinction accentuates the exotic spinon spectrum. I support the publication of this paper because of its importance, impact, and scope of interest, given that the authors present satisfactory responses to the points I present below.

I have three questions for the authors to modify or clarify.

First, the J derived from Zn-barlowite's Curie-Weiss temperature is likely larger than it actually is. This is due to a correction when extrapolating a mean-field theory down to temperatures comparable to and lower than J 's scale. In herbertsmithite, a Curie-Weiss temperature of 300 K gives a J of 17 meV. For Zn-barlowite, such a correction reduces J to 12~13 meV. Consequently, the spinon-pair dynamic extends to $\sim 4J$ which is reasonable considering known theoretical ranges. This correction also brings down the magnon peak energy in EuCu_3 for reconciliation as presented in the Methods section. Meanwhile, the DFT calculation in Section I of the SI results in a higher J than the one from mean-field theory. How do these reconcile?

Second, the authors almost shy away from talking about the spin gap in the main text but leave additional discussions to Section V of the SI. This is certainly a tough topic. Nevertheless, the presence or lack thereof a spin gap is at the core of this research field. In Figure 3e, the leveling at low temperatures indicates the lack of a spin gap. Direct comparison with ref[57] also hints at gapless spinons. That said, this interpretation is cautioned in Section V of the SI. Without loss of uncertainty, the authors ought to expand their discussion in the main text for this crucial information. In addition, there are excess Cu on the interlayer sites as it occurs in herbertsmithite. It would help to have more clarification for the roles these spins play.

Third, in Equation 1, why is the DM denoted along the z-axis?

List of main changes:

[Most changes in the main text are colored in Red.]

1. Page 1, an additional affiliation is added.
2. Page 2, more explanation is added about Eq. (1).
3. Page 6, the detail of the XX, XY and X-only configurations in the angle-resolved Raman is added.
4. Page 8, “The non-monotonic T -dependence deviating from the Curie-Weiss behavior is associated with the enhancement of nearest-neighbor spin correlations at low temperatures.[65]” is added.
5. Page 8 and 9, discussion about the spin gap is added.
6. Page 9, new figure (Fig. 4) for the power-law behavior for E_{2g} Raman continua at low frequency in Cu₃Zn is added.
7. The exchange strengths for Cu₃Zn and EuCu₃ are corrected to 13 meV and 7 meV, respectively. Thanks to the Reviewer #3 for this suggestion.
8. A new fund is added in the acknowledgement.
9. Other minor changes and typo corrections for presentation improvement in both main text and Supplementary Information.

REVIEWER COMMENTS

Reviewer #1 (Remarks to the Author):

The authors report a detailed Raman study of the magnetic excitations in QSL candidate $\text{Cu}_3\text{Zn}(\text{OH})_6\text{FBr}$ and compare it with the ordered antiferromagnet $\text{EuCu}_3(\text{OH})_6\text{Cl}_3$. A broad feature is seen in the Raman susceptibility of both, which eventually appears to become a 1 magnon excitation in the AFM ordered system. The structure is also evaluated using Raman selection rules. The paper is generally very well written and is a nice result comparing the two systems to potentially show how spinons become confined. I am generally inclined to recommend publication, however, the paper could polarizeruse some additional analysis to be at the level of nature communications. My specific comments are below:

[Reply] We thank the Reviewer for her/his nice summary and kind recommendation of our work. The point-to-point responses are given below.

1) The authors mention they use "X-only", but don't really explain why they chose to do so. It is not a common setup for Raman and thus it would be helpful to add a sentence. Similarly, was the incident or scattered light that was measured?

[Reply] We thank the Reviewer for her/his suggestion about the polarization setup for magnetic excitation measurements. The "X-only" configuration follows the theoretical proposal in Ref.39 [Cepas et al, PRB 77, 172406] to detect a weak symmetry breaking in the ground state, where the outgoing polarization is fixed and only incoming light is rotated. Actually, "X-only" polarization configuration is used in the previous study for the identification of lattice structure and lattice vibration [Chin.Phys.B 26(6), 067802 (2017); Phys. Rev. B 77, 172406 (2008); Nanoscale 7, 18708 (2015); Nat. Nanotechnol. 15, 212 (2020)]. For the lattice and spin dynamics with the threefold rotation symmetry, the Raman cross-section for the Raman modes belonging to irreducible representation E_g doesn't depend on the polarization direction of the incoming or outgoing light. However, it acquires a characteristic polarization dependence for a broken-symmetry state with an amplitude proportional to the emergent order parameter. Here, the "X-only" polarization configuration is used to confirm the threefold rotation symmetry of the lattice structure and spin dynamics of the $\text{Cu}_3\text{Zn}(\text{OH})_6\text{FBr}$ and $\text{EuCu}_3(\text{OH})_6\text{Cl}_3$ single crystals. Indeed, the Raman intensities of A_{1g} and E_g modes in the XX and XY configurations, as well as the E_g modes in X-only configuration in the two single crystals remain invariant at 290 K and 4 K (Fig.2), which should be an indicative of no symmetry breaking. Notably, we kept the outgoing polarization in order to avoid the varied diffraction efficiency of the grating for light with different polarization.

We thank the Reviewer for her/his suggestion and add a few words as following in the main text (Page 6, the second paragraph from bottom):

“In the XX (XY) configuration, the incoming and outgoing light polarizations are parallel (perpendicular) and rotated simultaneously. In the X-only configuration, the outgoing polarization is fixed and only incoming light is rotated. Theoretically, the Raman cross-section of a Kagome QSL ground state does not depend on the polarization of the incoming or outgoing light[39] and keeps invariant against rotating light polarization in the XX, XY, and X-only configurations.”

2) The authors mention observing a high T continuum in A1g that disappears upon cooling. Could this be the quasi-elastic scattering often seen in such systems? If so does its temperature dependence follow the magnetic susceptibility of the sample?

[Reply] The magnetic continuum in the A1g channel is the quasi-elastic scattering, related to the magnetic contribution to specific heat [please refer to Lemmens’s review article Ref.41], not the magnetic susceptibility.

As we discussed in the manuscript, A1g channel measures the thermal fluctuations in Cu₃Zn and the corresponding integrated Raman susceptibility is proportional to the energy fluctuations of the kagome spins $\chi'(T) \propto \langle (\delta E)^2 \rangle$ as the A1g Raman tensor is nothing but the Hamiltonian itself whose eigenvalues are the energy levels. The basic thermodynamics tells us that the heat capacity also measures the energy fluctuations $C(T) \propto \frac{\langle (\delta E)^2 \rangle}{T^2}$.

When the Raman tensor contains the single-spin-flip process in the spin-orbit coupling spin system, the integrated Raman susceptibility is related to the magnetic susceptibility of the sample, for example, Fig. 7 in Glamazda et al PRB 95, 174429 (2017) for α -RuCl₃.

We notice that in our measurements (Fig.3a), the signal-to-noise ratio for the Raman signal in the A1g channel is poor particularly below 90 K. Thus a serious comparison between the integrated and the heat capacity is not available at the current stage and also beyond the scope of the present project.

Fortunately, the quantum fluctuations of the kagome spins in the E2g channel persist down to 4 K (Fig.3d) with a good signal-to-noise ratio, which is benefit for us to figure out the spinon-antispinon excitation.

We think that the Reviewer may still be interested in the rough comparison between the experimental A1g Raman susceptibility and theoretical heat capacity. So we present the comparison below in Fig.R1. However, we will not present the result in the manuscript to avoid the over interpretation of the experimental data.

Figure R1. Rough comparison between the experimental AlG Raman susceptibility χ' and theoretical heat capacity. Left: We divide χ' by T^2 . Notice that the signal-to-noise ratio below 100 K is poor and the increase upon cooling is not reliable. The solid line is the thermal activated function divided by T^2 . Right: the theoretical heat capacity for the kagome antiferromagnetic model with various DM interactions (lines in different colors) [From Fig. 4a in PRL 125, 027203 (2020)].

3) In the Cu₃Zn the authors argue they observe the 1 pair and 2 pair spinon excitation. However, it is not really clear how they exclude 2 magnon excitations? In other words, do they have direct evidence these are from fractional excitations? For example, there is the recent work by the Burch group showing how to analyze the continuum to demonstrate the statistics are fermionic, have the authors done something similar or directly compared to theoretical calculations? (Note the authors do not cite the latest works on RuCl₃ from Choi and Burch, the former shows magnons emerging from the continuum in a magnetic field and the later how to analyze the spinon continuum).

[Reply] This is one of the most important issues in the magnetic Raman continuum which has been frequently used as evidence for spinon excitations in QSL. As we stressed in our manuscript, the Raman continuum doesn't necessarily imply the spinon. Since two-magnon excitations in AFM order can also give rise to the Raman continuum, we need more analysis to connect the magnetic continuum with spinon excitations in our study as follows:

a. Our Raman profile agrees with the theory of kagome QSL and can be decomposed into one-pair and two-pair components of spinon-antispinon excitations. The maxima and cutoffs observed in Raman spectroscopy agree well with the theoretical prediction. While two-pair resembles two-magnon, the one-pair continuum comes from the spinon excitation. Furthermore, the one-pair continuum data are found to exhibit the power-law behavior, consistent with the theory for Raman intensity profile by spinon-antispinon pairs [PRB 81, 024414(2010)].

b. From the control experiment (Fig. 6 in the revised manuscript), we can see that the maximum of the one-pair continuum has the same energy scale as the magnon peak in EuCu₃(OH)₆Cl₃. In addition, the distinct difference between magnon peak of EuCu₃ and Raman continuum in Cu₃Zn in the same energy scale provides strong evidence for the one-pair spinon excitation of Raman continuum in Cu₃Zn. Accordingly, the two-pair spinon excitation rather than the two-magnon excitation make

contributions to the observed Raman continuum of Cu_3Zn in the range of $50\text{-}750\text{ cm}^{-1}$ with the maximum at 400 cm^{-1} .

c. Compared with the inelastic neutron experiments, the one-pair Raman continuum of Cu_3Zn observed in our Raman spectroscopy extends in the same energy range as the neutron continuum in Herbertsmithite.

We thank the Reviewer for reminding us of the work on the Raman continuum in $\alpha\text{-RuCl}_3$. We cite the papers from Choi [Ref.49] and Burch [Ref. 50] in the revised manuscript. Thanks to the exact solvability of the Kitaev model, the fermionic statistics can be derived in $\alpha\text{-RuCl}_3$ from the direct comparison of the temperature dependence in the Raman intensity. Unfortunately, the Kagome antiferromagnetic model has no analytical or reliable numerical result for spin dynamics even at zero temperature and a direct quantitative comparison between experiments and theory is not available.

Reviewer #2 (Remarks to the Author):

In this manuscript, the authors report Raman scattering data on single crystals of two quantum kagome antiferromagnets, of which one is the kagome QSL candidate $\text{Cu}_3\text{Zn}(\text{OH})_6\text{FBr}$ (short notation Cu_3Zn), and another is an antiferromagnetically ordered compound $\text{EuCu}_3(\text{OH})_6\text{Cl}_3$ (short notation EuCu_3).

The authors estimated the superexchange strength for the kagome spins in Cu_3Zn as $J=19\text{meV}$ and Curie-Weiss temperature $\Theta_{\text{CW}}=-220\text{K}$. The superexchange interaction for EuCu_3 is about $J=10\text{meV}$. They also established the structurally ideal realization of the kagome lattice of Cu_3Zn by SHG and angle-resolved Raman scattering.

At high temperatures, the two compounds show qualitatively the same Raman intensity, but at low temperatures the response is very different. A sharp magnon peak appears in EuCu_3 in the E_g magnetic Raman continuum below the magnetic transition temperature $T_N=17\text{K}$. In Cu_3Zn , at high temperature, the authors observe continuum in both E_{2g} and A_{1g} symmetry channels.

At low temperatures, only E_{2g} response remains, i.e. the authors see a remarkable E_{2g} magnetic Raman continuum, which differently from the response in A_{1g} channel persists down to 4 K, indicating the intrinsic quantum fluctuation of the kagome spins. No sharp peaks at low T is observed in Cu_3Zn , and a beautiful comparison of the Raman response in two compounds is presented in Fig.5. Then the authors claimed that it can be decomposed into one-pair and two-pair components of spinon excitations, in agreement with previous theoretical works.

These findings are interesting and deserve publishing.

[Reply] We thank the Reviewer for the comprehensive summary and the positive comment of our work.

However, I have some doubts about the interpretation of the experimental data. The authors claimed that it can be decomposed into one-pair and two-pair components of spinon excitations, in agreement with previous theoretical works. They also discuss the implication of the power-law behavior in the Raman susceptibility in the E_{2g} channel at low frequency and suggest that the spinons might correspond to those in the Dirac kagome QSL. However, if a theory for the Raman response in the Dirac kagome QSL exists, why not to present it along with the experimental data, or at least perform a qualitative comparison? Also it is not clear for me if this theory can account for strongly non-monotonic T-dependence. The authors do not discuss this.

[Reply] Indeed there are such theoretical studies. Ten years ago, one of us (PA Lee) theoretically studied the Raman intensity of the kagome QSL at zero temperature [PRL 98, 117205 (2007), PRB 81, 024414 (2010)], serving as the spectroscopic signature of spinon excitations. It predicted that the E_g magnetic Raman continuum contains one- and two-pair spinon excitation components as shown in Fig.R2, which is in agreement with our experimental results observed by Raman spectroscopy.

Figure R2: The theoretical E_g Raman intensity for the Dirac kagome QSL [Fig.6 in PRB 81, 024414, 2010]. In the theory, the intensity is calculated on the mean-field level. Comparing to the experimental results in Fig.3d in our present work, the two-pair component is overestimated in the mean-field theory, probably due to the correlation effect which is not well treated on the mean-field level.

The non-monotonic T-dependence is quite general in the spin system and can be qualitatively explained as follows:

At high temperature, the spin fluctuation follows the Curie-Weiss behavior, increasing with the decreasing temperature, as the correlations are overwhelmed by the thermal fluctuations. At low temperatures, the spin correlation due to the strong interactions suppresses the fluctuations when further reducing temperature. As an analogy, in the famous high- T_c cuprate superconductors, the maximum temperature of the non-monotonic T-dependence of the magnetic susceptibility is used to be defined as the pseudogap temperature [Doping a Mott insulator: Physics of high-temperature superconductivity, PA Lee, N. Nagaosa, and X.G. Wen, RMP, 78, 17 (2006)].

The non-monotonic T-dependence of the related physical properties, e.g. specific heat and magnetic susceptibility, of the kagome antiferromagnet are also found theoretically, e.g., in a recent theoretical paper, Bernu et al, PRB 101, 140403, 2020.

We thank Reviewer for her/his suggest and add a few words in the revised manuscript (Page 9, the paragraph above Fig. 4):

“The non-monotonic T-dependence deviating from the Curie-Weiss behavior is associated with the enhancement of nearest-neighbor spin correlations at low temperatures.[65]”

Also, in EuCu_3 , the authors claimed that the magnon Raman peak is a direct spectroscopic evidence for the $q=0$ 120 degree non-collinear AFM spin configurations, but no comparison with theoretical results even on the level of the linear spin wave theory is presented.

[Reply] The linear spin-wave theory is presented in the method section, see Eqs. (8) and (9).

I think the reader will benefit a lot if the authors address some of these questions prior to publication.

[Reply] We thank the Reviewer again for her/his suggestions.

Reviewer #3 (Remarks to the Author):

The search for experimental realizations of quantum spin liquids has long been a daunting challenge. Kagome systems bear utmost promises with herbertsmithite $\text{ZnCu}_3(\text{OH})_6\text{Cl}_2$ being in the limelight for a decade. That said, circumstantial evidence casts looming doubts on its ground state's characteristics. Attempts at replacing Zn ions did not quench caveats. In the past few years, barlowite and its derivatives paved a way to replace the Cl in herbertsmithite. Strong frustration in both materials amplifies the effect of minimal perturbations in their structures, such that stepping away from herbertsmithite yet remaining in the spin-liquid regime is all the more desirable. Pioneer studies on Zn-barlowite show strong support of a spin-liquid ground state. More direct observations of spinons are obviously the imperatives.

The authors show spinon-pair excitation using Raman spectroscopy conducted on a single crystal sample. Similar observation was one of the inspiring evidence in herbertsmithite for which slight structural distortion has been reported recently. The authors carefully demonstrate the robustness of structure in Zn-barlowite, subject to the resolution of Raman spectroscopy, before presenting magnetic excitation which resemble those observed in herbertsmithite. The spectral profile is consistent with theoretical predictions on a Dirac $U(1)$ gapless spin-liquid state overall. In addition, the availability of $\text{EuCu}_3(\text{OH})_6\text{Cl}_3$ ---a resembling compound with an antiferromagnetic order below 17K---allows the authors to contrast magnon pairs vs spinon pairs. The distinction accentuates the exotic spinon spectrum. I support the publication of this paper because of its importance, impact, and scope of interest, given that the authors present satisfactory responses to the points I present below.

[Reply] We thank the Reviewer for the summary of our work and the recommendation of the publication.

I have three questions for the authors to modify or clarify.

First, the J derived from Zn-barlowite's Curie-Weiss temperature is likely larger than it actually is. This is due to a correction when extrapolating a mean-field theory down to temperatures comparable to and lower than J 's scale. In herbertsmithite, a Curie-Weiss temperature of 300 K gives a J of 17 meV. For Zn-barlowite, such a correction reduces J to 12~13 meV. Consequently, the spinon-pair dynamic extends to $\sim 4J$ which is reasonable considering known theoretical ranges. This correction also brings down the magnon peak energy in EuCu_3 for reconciliation as presented in the Methods section. Meanwhile, the DFT calculation in Section I of the SI results in a higher J than the one from mean-field theory. How do these reconcile?

[Reply] We thank the Reviewer for raising this issue. We simply use the Curie-Weiss temperature to estimate the interaction in our manuscript. In the revised manuscript, we use the reduced ones with some corrections in the pre-factor of our energy scales. The interactions have the modified values \$J\$ of 13 meV and 7 meV for \$\text{Cu}_3\text{Zn}\$ and \$\text{EuCu}_3\$, respectively. The correction doesn't change the main conclusions in this work.

The DFT estimation of the interaction J depends on the value of “ U ” in the DFT+ U scheme. Roughly speaking, $J = \frac{4t^2}{U}$ as in the one-band Hubbard model for spins in 3d9 configuration of Cu^{2+} . In the DFT simulation, the kinetic energy is well treated and the hopping parameter t is reliable. However, the strong correlation of the Hubbard physics can’t be well treated in the DFT simulation. To account for the strongly correlated physics, usually an empirical value of “ U ” is added in the energy functional in the DFT+ U scheme. How to choose a “good” U value is tricky, but a value of 5-10 eV is reasonable. Note the value of U also depends on the functional, e.g., LDA, PBE or SCAN, that we choose in DFT simulation, and on whether we relax the crystal structures before we estimate J .

In our calculation, we take $U = 6$ eV and obtain reasonable estimations of J although they are higher than the values from mean-field theory. If we use a larger “ U ” value, we have a smaller J . We didn’t expect exact values out of the DFT estimations, and hence didn’t finely tune the “ U ” in the DFT+ U calculation. The main information from the DFT simulation is the ratio of D/J , weakly depending on the value of U , which explains the reason why Cu_3Zn and EuCu_3 have different ground states.

Second, the authors almost shy away from talking about the spin gap in the main text but leave additional discussions to Section V of the SI. This is certainly a tough topic. Nevertheless, the presence or lack thereof a spin gap is at the core of this research field. In Figure 3e, the leveling at low temperatures indicates the lack of a spin gap. Direct comparison with ref[57] also hints at gapless spinons. That said, this interpretation is cautioned in Section V of the SI. Without loss of uncertainty, the authors ought to expand their discussion in the main text for this crucial information. In addition, there are excess Cu on the interlayer sites as it occurs in herbertsmithite. It would help to have more clarification for the roles these spins play.

[Reply] We thank the Reviewer for the insightful suggestions. We move the discussion about the spin from SI to the main text with some expansion (Pages 8 & 9 in the revised manuscript).

In the previous manuscript, we have discussed the interlayer Cu^{2+} with the concentration (18%) in Cu_3Zn for the disorder effect in the magnetic Raman scattering. The Raman scattering measures the nearest neighbor spin-pair dynamics, but the spin pairs associated with the interlayer Cu^{2+} ions are weaker than the singlet pairs for the kagome spins:

- (a) The spin-pair between impurities is small;
- (b) The spin-pair between impurity and kagome spin is small.

As a result, the interlayer Cu^{2+} ions contribute a negligible Raman matrix element and we can safely ignore their effect in the discussions about the Raman experiments.

Third, in Equation 1, why is the DM denoted along the z-axis?

[Reply] From the EPR results in Herbertsmithite [Zorko et al, PRL 101, 026405, 2008] and $\text{YCu}_3(\text{OH})_6\text{Cl}_3$ [Arh, PRL 125, 027203, 2020], we know that the in-plane DM interaction is much smaller than that along the z-axis. Zn-Barlowite and $\text{EuCu}_3(\text{OH})_6\text{Cl}_3$ have similar local chemistry

environments for Cu-O-Cu bonds in the kagome planes, and hence we assume only the DM interaction along the z-axis.

We thank the Reviewer for her/his suggestion and add a few words in the manuscript (page 3, below Eq. (1)):

“...We ignore the in-plane DM interactions regarding to the previous electron paramagnetic resonance measurements in the related kagome systems. [55, 56]”

REVIEWERS' COMMENTS

Reviewer #1 (Remarks to the Author):

The authors have done a reasonable job of answering the questions raised by all referees and the paper is now suitable for publication in NAture Communications.

Reviewer #2 (Remarks to the Author):

I am happy with the authors reply and recommend the manuscript for publication.

Reviewer #3 (Remarks to the Author):

The authors have satisfactorily addressed my concerns. I recommend publication.